# Removal of Amoxicillin from Processing Wastewater by Ozonation and UV-Aided Ozonation: Kinetic and Economic Comparative Study

Beatriz Santos Silva [1], Mariana Cardoso Barros Ribeiro [1], Bruno Ramos [2] and André Luís de Castro Peixoto [1,*]

1   Research Group on Process Chemistry (ProChem), Federal Institute of Education, Science and Technology of Sao Paulo (IFSP), Capivari Campus, Capivari CEP 13365-010, SP, Brazil
2   Research Group on Advanced Oxidation Processes (AdOx), Department of Chemical Engineering, Polytechnic School, University of São Paulo, Sao Paulo CEP 05508-000, SP, Brazil
*   Correspondence: alcpeixoto@ifsp.edu.br

**Abstract:** This work presents an empirical and scaling-up study of the degradation and mineralization of amoxicillin (AMX) from expired pharmaceutical formulations by $O_3$-based processes. A set of UV–ozone-based experiments was used to model the kinetics of AMX degradation, considering several chemical/photochemical mechanisms (hydrolysis, direct ozonation, radical reactions, and photolysis). Finally, the modeling data were used for scaling-up purposes, considering CAPEX and OPEX costs on the US Gulf Coast basis. In terms of experimental results, the amoxicillin (AMX) pharmaceutical effluent was successfully degraded by ozone technology at high pH values. The semi-batch ozonation process was effective after 60 min of treatment in all experimental conditions, producing degradation intermediates recalcitrant to $O_3$ oxidative process. From the bench-scale kinetics, scaling-up simulations indicate that the gain provided by adding a UV unit does not compensate for the increase in capital and operational costs of adding irradiation equipment. It suggests ozonation at high pH as the best cost-effective approach to degrade AMX. The figures-of-merit electric energy consumption per order ($E_{EO}$) corroborates the scaling-up simulations. $E_{EO}$ results indicate no-UV ozonation as the best option to degrade AMX at high pH values. The $E_{EO}$ of the present work showed a lower energy consumption system than previous papers from the literature.

**Keywords:** amoxicillin; ozone; advanced oxidation processes; scale up; CAPEX; OPEX; data modeling; electric energy consumption per order ($E_{EO}$)



## 1. Introduction

Amoxicillin ((2S,5R,6R)-6-[[(2R)-2-amino-2-(4-hydroxyphenyl)acetyl]amino]-3,3-dimethyl-7-oxo-4-thia-1-azabicyclo [3.2.0]heptane-2-carboxylic acid), with chemical formula $C_{16}H_{19}N_3O_5S$, as shown in Figure 1, is an antibiotic that belongs to the beta-lactams group and is often used for bacterial infection treatments [1]. The structure of amoxicillin exhibits amphoteric properties due to three main functional groups, $-NH_2$, $-COOH$, and $-OH$ [2]. Because of this diversity, amoxicillin has three different acid dissociation constants, namely $pKa_1 = 2.68$ (carboxyl group), $pKa_2 = 7.49$ (amine group), and $pKa_3 = 9.63$ (phenol group) [3]. Thus, amoxicillin can exist in aqueous solution in four different forms (i.e., $AMX^+$, $AMX$, $AMX^-$, and $AMX^{2-}$) depending on the pH of solution, as shown in Figure 2.

In Brazil, amoxicillin is part of the National Policy of Pharmaceutical Assistance (NPPA) of the Ministry of Health. Amoxicillin is, therefore, one of the essential medicines supplied free of charge throughout the national territory as a public policy. According to the World Health Organization [4], the overall consumption of antibiotics ranged from 4.4 to 64.4 Defined Daily Doses (DDD) per 1000 inhabitants per day. In most countries (including Brazil), amoxicillin and amoxicillin/clavulanic acid are the most frequently consumed antibiotics. Antibiotic consumption in Brazil is approximately 22.75 DDD, higher than

other countries in the Americas such as Bolivia (19.57), Paraguay (19.38), Canada (17.05), Costa Rica (14.18), and Peru (10.26) [4,5].

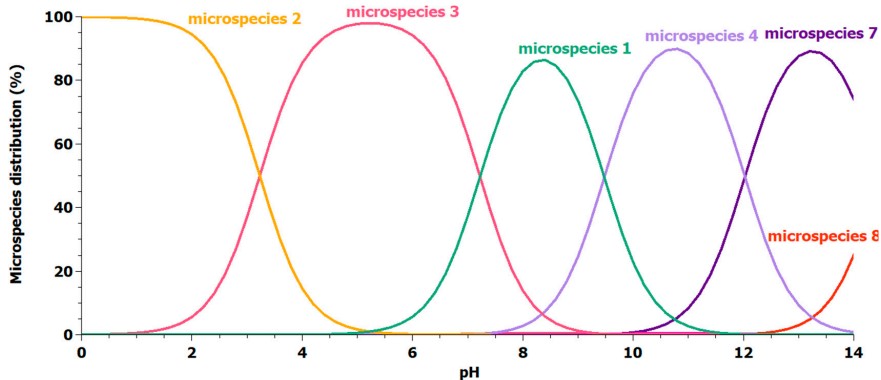

**Figure 1.** AMX Chemical structure.

**Figure 2.** Speciation of AMX as a function of pH in aqueous solutions. The chemical structure of each microspecies is shown in the Supplementary Materials (Figures S1–S8).

The use of antibiotics in animals is nearly triple the human consumption. In 2013, the global use of antibiotics in livestock was 131,109 tons, and the projection for 2030 rises above 200,000 tons. Furthermore, consumption levels vary between countries. In Norway, it reaches 8 mg per kilogram of animal product, whereas in China, the level can be up to 318 mg per kilogram of animal product [6].

Despite the importance of the consumption of antibiotics such as amoxicillin, our excessive use and indiscriminate disposal into wastewater streams have been causing major contamination in the ecosystem, which contributes to the development of antibiotic-resistant pathogens [7,8]. The presence of AMX in surface water, groundwater, and industrial and domestic wastewater is well-documented [2]. Part of this contamination is explained by the irregular disposal of expired pharmaceutical formulations, which, in many countries, are simply discarded into the toilet or as common household waste. Although there are no official data, it is estimated that, in 2015, 90% of 34.8 billion daily doses of antibiotics ended up in the environment as active substances [9].

Antibiotics and other recalcitrant substances are organic substances with low biodegradability. For such chemical compounds, their removal by biodegradation processes is not effective [10]. On the other hand, several studies demonstrated the efficiency of advanced oxidation processes (AOPs) in the degradation of a wide spectrum of organic pollutants [11–15], treating highly stable contaminants. AOPs are designed to enhance the degradation and mineralization of pollutants or to transform them into less toxic compounds [16]. The mechanism of action of AOPs includes the production of hydroxyl radicals ($^{\bullet}$OH), a potent oxidizing agent ($E^{\circ} = 2.8$ V) capable of interacting with a wide range of organic compounds. $^{\bullet}$OH can be produced by oxidizing agents such as ozone ($O_3$) [17] or hydrogen peroxide ($H_2O_2$) [18] as well as from the photoexcitation of semiconductor/metal oxides [19,20].

Ozone-based technologies can promote the degradation of pharmaceutical compounds by attacking unsaturated hydrocarbon bonds in their structure. The combination of ozone and UV light in aqueous media induces the formation of highly oxidative radicals ($^\bullet$OH and HO$_2^\bullet$) which enhance he performance of the oxidation process. Moreover, in alkaline conditions, more O$_3$ may be transformed into $^\bullet$OH, and indirect oxidation of pharmaceutical compounds is more beneficial in the degradation process than direct oxidation by O$_3$ [21]. The following equations illustrate the O$_3$ decomposition reaction (indirect oxidation mechanisms):

$$O_3 + OH^- \rightarrow HO_4^- \tag{1}$$

$$HO_4^- \leftrightarrow HO_2^\bullet + O_2^{-\bullet} \tag{2}$$

The products HO$_2^\bullet$ e O$_2^{-\bullet}$ will be disproportionate in the absence of molecular ozone to form hydroxyl $\left( HO^{2-} \right)$ and oxygen gas (O$_2$).

Subsequently, hydroxyl and oxygen gas quickly protonate to hydrogen peroxide. The absence of ozone is illustrated by the equations below:

$$O_2^{-\bullet} + O_3 \rightarrow O_2 + O_3^{-\bullet} \tag{3}$$

$$O_3^{-\bullet} \rightarrow O_2 + O^{-\bullet} \tag{4}$$

$$O^{-\bullet} + H_2O \rightarrow \bullet OH + OH \tag{5}$$

In this work, we investigated the application of O$_3$ at high pH, with and without UV–C irradiation, for the degradation and mineralization of AMX in water in a gas–liquid biphasic reactor system. The AMX formulation used in this investigation was provided to us with expired shelf life and from a public pharmacy (acquired within the NPPA). Kinetics modeling, reactor scaling-up, and economic considerations complete the ozone-based AOP experimental study, providing novelties to the AOP scientific literature with a complex residue degradation process. Our work is justified in this context of increased consumption of amoxicillin and significant generation of pharmaceutical waste. The present work was designed to be implemented by public and/or private sectors, and the economic analysis is part of the strategy.

## 2. Material and Methods

### 2.1. Material

The water used for chemical analyses and the chemical processes was purified with a Merck Milli-Q Direct 8 System. All reagents were provided at analytical grade (Sigma-Aldrich, Saint Louis, MO, USA). The AMX pharmaceutical formulation (Prati Donaduzzi, Toledo, Brazil) used to prepare the simulated wastewater treatment plant (WWTP) effluent expired in January 2019. Pharmaceutical formulation in powder for suspension with a nominal concentration of 250 mg AMX/5 mL was used to prepare the simulated WWTP effluent. The pharmaceutical formulation used in the ozone-based experiments had no potassium clavulanate. The simulated wastewater was prepared by solubilization of 1.5000 g of AMX pharmaceutical formulation in 3.00 L calibrated flask, completing the volume with pure water (Type III water, ASTM D1193-91). The AMX calibration curve was obtained with ultrapure amoxicillin trihydrate (Ventranal®, Sigma-Aldrich). All chemical analyses were carried out in ultrapure (Type I) water, except for the spectrophotometric analyses of AMX, where 1-propanol (Sigma-Aldrich, purity $\geq$ 99.5%) was used as solvent.

### 2.2. Methods

#### 2.2.1. Chemical Analyses

AMX Determination. The analysis of AMX concentration was performed according to the methodology described by Saleh [22]. More details of the methodology were described in our previous paper [23]. For the analyses, the laboratory was kept at a temperature of 20 °C. The samples had their pH values corrected to ~7 promptly after being taken from

the reactor. In a 10.00 mL calibrated flask, 1.00 mL of sample, 1.00 mL of 0.1 mM NaOH (Sigma-Aldrich, purity $\geq$ 99.0%), 1.00 mL of 5% N-Chlorosuccinimide (NCS, Sigma-Aldrich, purity $\geq$ 98%), and enough 1-propanol to fulfill the flask were added in sequence. The absorbance was measured at 395 nm against a blank solution in a Genesys 10S UV-Vis spectrophotometer (Thermo Fisher Scientific, Waltham, MA, USA). The blank was carried out with all reagents except AMX. The samples were read promptly at their preparation to avoid errors due to solvent volatility.

Total Organic Carbon and Total Nitrogen. Total Organic Carbon (TOC) analyses were performed according to the combustion catalytic oxidation method [24] under an $O_2$ atmosphere (IBG synthetic air, Jundiaí, Brazil, purity $\geq$ 99.9995%). The analyses were performed with a non-dispersive infrared detector (NDIR). Total Nitrogen (TN) analyses were performed according to the chemiluminescence method [25] in a TNM-L Shimadzu (Kyoto, Japan) equipment coupled with the TOC-L apparatus (Shimadzu, Kyoto, Japan). The TN chemiluminescence method does not differentiate inorganic and organic forms of nitrogen. Non-Purgeable Organic Carbon (NPOC) was carried out, pretreating the samples with HCl solution followed by injection of ultrapure synthetic air before the sample injection into the high-temperature oven (>680 °C). HCl solution was prepared with a 36.5% $w/w$ hydrochloric acid solution (Vetec, Duque de Caxias, Brazil).

### 2.2.2. AMX Degradation Processes

Three-liter amoxicillin pharmaceutical formulation wastewater was ozonated in a Polyvinyl Chloride (PVC) tank column. The ozonation system was carried out in a semi-batch mode wherein the $O_3$ +$O_2$ was continuously sparged at a rate of up to 1.0 L min$^{-1}$. The gas was sparged through a sintered glass dispersion cylinder with a dimension of 15 $\times$ 20 mm. A corona discharge bench-scale ozone generator (Ozone&Life, model O&L3.0 RM, São José dos Campos, Brazil) produced ozone with a maximum capacity of 3.0 g h$^{-1}$ $O_3$. The $O_3$ generator was fed with oxygen gas (White Martins, Jundiaí, Brazil, purity $\geq$ 99.0%). The reaction system temperature was set at (20.0 $\pm$ 0.5) °C (thermostatic bath by SL-152 Solab equipment, Piracicaba, Brazil). NaOH and $H_2SO_4$ aqueous solutions were used to adjust the pH in the reaction media. The effluent was recirculated in the system for a maximum of 120 min, and sample aliquots were collected at predetermined intervals. The chemical analyses were necessary to verify the degradation yield of the ozonation-based process and evaluate kinetics data for the economical computational simulation. No buffer was used to control the reaction pH, only NaOH and $H_2SO_4$ (substances both of which cause no UV interference in the reaction media). Figure 3 shows the experimental setup. The photolytic experiments were performed with two low-pressure mercury lamps (Osram, Munich, Germany, 15 watts each lamp). The 30 watt UV system was monochromatic with wavelength ($\lambda$) of 253.7 nm.

### 2.2.3. Kinetic Data and Economic Analysis

The experimentally measured AMX concentration data were fit numerically to a pseudo-first-order model using nonlinear least-squares (LSQ) optimization, considering the experimental arrangement shown in Figure 3 using data analysis software (Origin 2020, OriginLab Co., Northhampton, MA, USA). Experimental data on the concentration of AMX over time for different reaction conditions (e.g., with ozone feed only, with irradiation only, and with both ozone supply and UV irradiation, pH) were used to extract the relevant apparent kinetic parameters. Simulations using extracted kinetic data were carried out in a numerical software package (Matlab 2021a, Mathworks Inc., Natick, MA, USA). The comparison between the two processes (ozonation and Ozone/UV at pH 9.00, 11.00, and 13.00) was based on performance evaluation and economic considerations. Equipment sizing was evaluated for three different plant operation regimes: one-, two- and three-shift labor systems. For estimation purposes, each shift was considered an 8 h day carried out by a single operator. Thus, in our scenarios, operating in two- or three-shift systems requires adding one or two additional plant operators. Additionally, considering the bench

system's typical conversion efficiency, our scenarios simulated the possibilities of carrying out the shift load in one, two, or three batches. According to typical industrial targets and legislation requirements, a target conversion of 80% of the initial antibiotic concentration was set [26]. A contaminant conversion of 80% is a minimum requirement of Article 18 of the State of São Paulo Environmental Company (CETESB) for the direct or indirect release of an effluent into water bodies. The target processing throughput was fixed at 10 m$^3$ per day within small-scale pharmaceutical processing plants [27]. Capital and operational costs were estimated according to suitable estimation methodologies developed for a US plant scenario [28]. Labor costs were calculated from official US data [29], and the remaining fixed costs were estimated using factors from ISBL, OSBL, and labor costs, wherever appropriate.

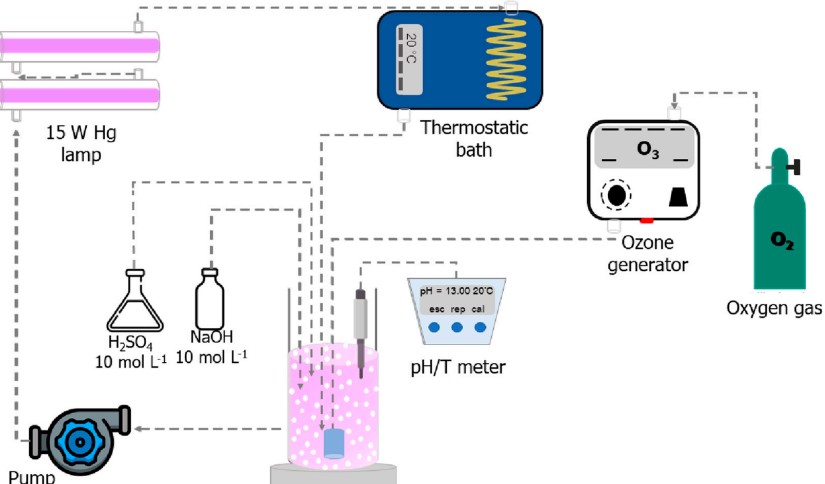

**Figure 3.** Semi-batch ozone-based advanced oxidation process diagram for the degradation of AMX in pharmaceutical formulation wastewater.

For low contaminant concentrations, where the overall kinetics is well represented by pseudo-first order models, the *electric energy consumption per order* ($E_{EO}$) is a preferable figure-of-merit for comparison of electricity-driven degradation processes [30,31]. $E_{EO}$ is also relevant as a parameter of economic viability study. $E_{EO}$ is defined as the electric energy in kilowatt hours (kWh) required to degrade a contaminant by one order of magnitude in a unit volume (e.g., 1 m$^3$) of contaminated water. $E_{EO}$ values (kWh m$^{-3}$ order$^{-1}$) were calculated using the Equation (6) [30,32]:

$$E_{EO} = \frac{38.4P}{Vk} \tag{6}$$

where $P$ was the rated power (kW), $V$ the reaction volume (L), and $k$ the first-order apparent degradation rate constant (min$^{-1}$). The ozone generator rated power was 80 W, and the UV lamps, 30 W. In O$_3$/UV processes, the rated power was the energy consumption by the lamps plus the ozone generator. In all cases, the reaction volume was 3.0 L. One-order of magnitude means an AMX degradation of 90% (C/C$_0$ = 0.10).

## 3. Results

### 3.1. Characterization of [AMX]$_0$ and TOC$_0$ as a Function of the Mass of the Pharmaceutical Formulation

Since a study involved real residues of pharmaceutical formulation with expired shelf life, it was decided to evaluate the initial concentration of AMX, [AMX]$_0$, and of total organic carbon (TOC), TOC$_0$, with respect to the mass of the pharmaceutical formulation. Figure 4 shows a box plot chart (*n* = 25 samples).

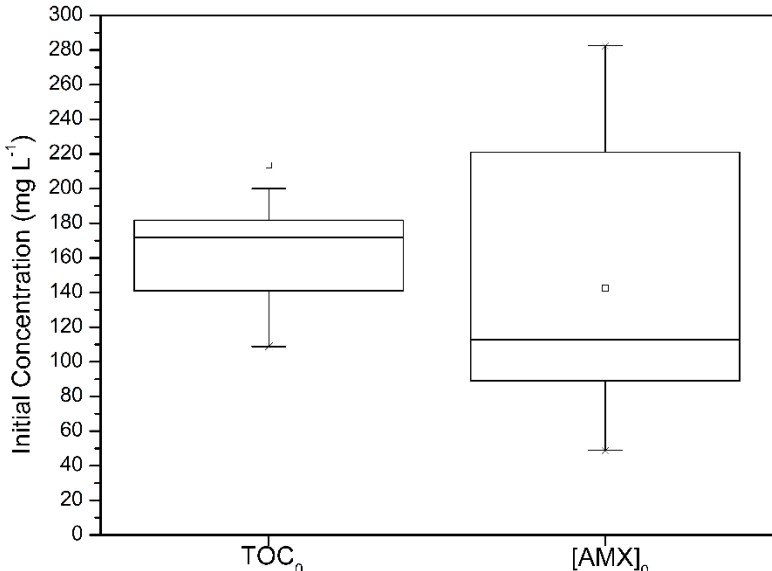

**Figure 4.** Boxplot of $[AMX]_0$ and $[TOC]_0$ as a function of the mass of pharmaceutical formulation (sample size, *n* = 25). Pharmaceutical formulation mass of $(1.5049 \pm 0.0053)$ g. (mean $\pm$ standard deviation). Reaction volume = 3.00 L.

### 3.2. Experiments of Hydrolysis in Alkaline Media

As the experiments of AMX degradation were studied in high pH, a second step of the project was to evaluate a possible mechanism of hydrolysis during each AMX degradation experiment. Figure 5 presents the normalized results of AMX concentration profile ($C/C_0$) observed in 120 min. Each experimental condition was carried out in triplicate, and the temperature was set at $(20.0 \pm 0.5)$ °C. The pH values were adjusted by NaOH in the same way as in ozone-based advanced oxidation experiments. When necessary, $H_2SO_4$ was also used to adjust pH. The amount of NaOH or $H_2SO_4$ did not interfere with the AMX concentration.

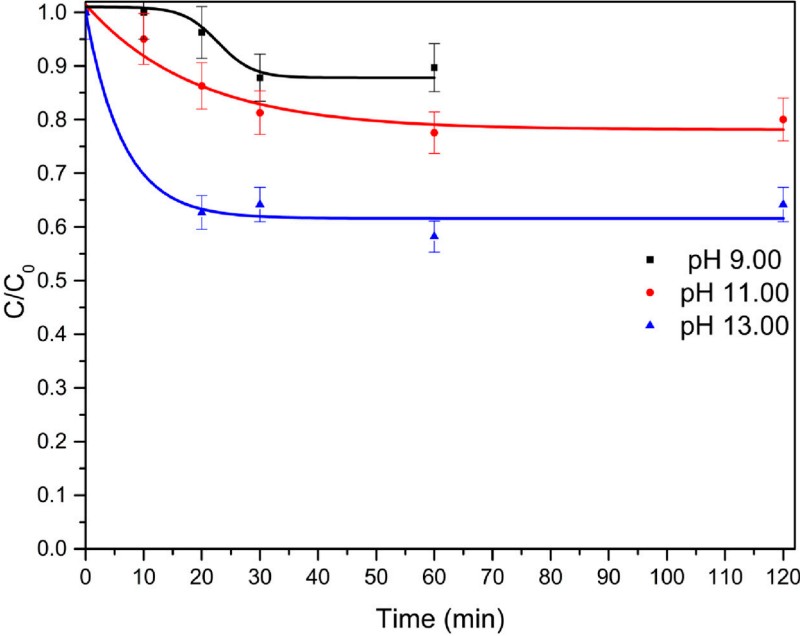

**Figure 5.** AMX hydrolysis in alkaline media. Results based on normalized concentration ($C/C_0$). Experiments performed in triplicates at pH 9.00, 11.00, and 13.00. Temperature control at $(20.0 \pm 0.5)$ °C.

### 3.3. Experiments of Ozonation in Alkaline Media

Figures 6 and 7 show the results of AMX degradation by ozonation-based technology at pH 9.00 and 11.00, respectively. It adopted two ozone feeding conditions: 8.13 mg min$^{-1}$ and 15.00 mg min$^{-1}$. The ozone mass–rate condition was kept constant for each experiment. Temperature and pH were also kept constant for each experiment.

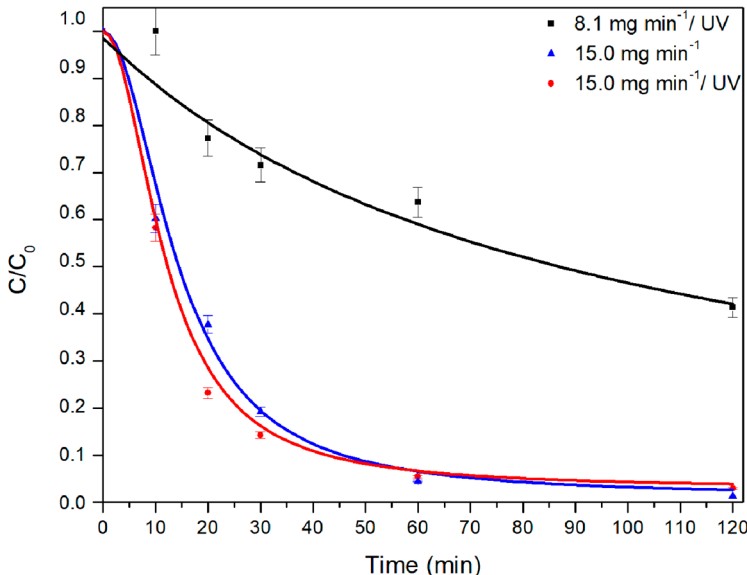

**Figure 6.** AMX degradation (C/C$_0$) by ozone-based advanced oxidation processes. Experiments carried out at pH 9.00 and temperature control at (20.0 ± 0.5) °C. Feed ozone mass–rate: 8.13 mg min$^{-1}$ and 15.00 mg min$^{-1}$.

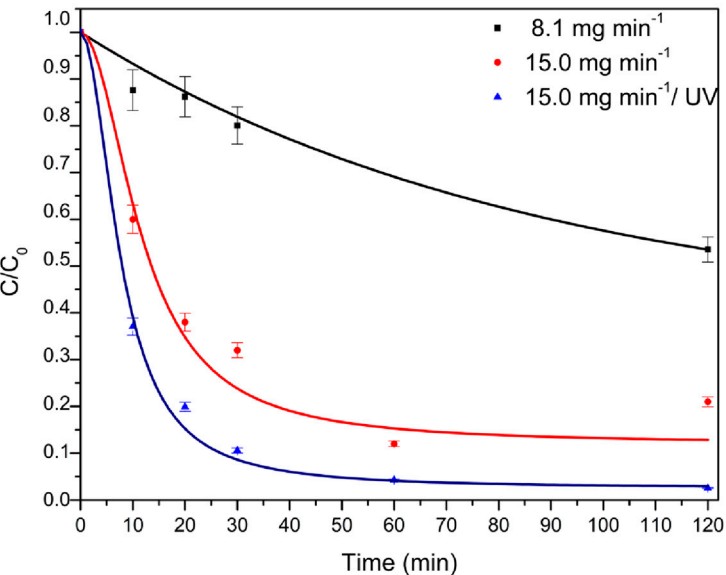

**Figure 7.** Experiments on AMX degradation (C/C$_0$) by ozone-based advanced oxidation processes. Experiments carried out at pH 11.00 and temperature control at (20.0 ± 0.5) °C. Feed ozone mass–rate: 8.13 mg min$^{-1}$ and 15.00 mg min$^{-1}$.

Figure 8a,b show the results of AMX degradation by ozonation-based technology at pH 13.00. Five ozone feeding conditions were adopted: 5.00 mg min$^{-1}$, 8.13 mg min$^{-1}$, 13.00 mg min$^{-1}$, 15.00 mg min$^{-1}$, and 25.00 mg min$^{-1}$. The ozone mass–rate condition was kept constant for each experiment. Temperature and pH were also kept constant for each experiment.

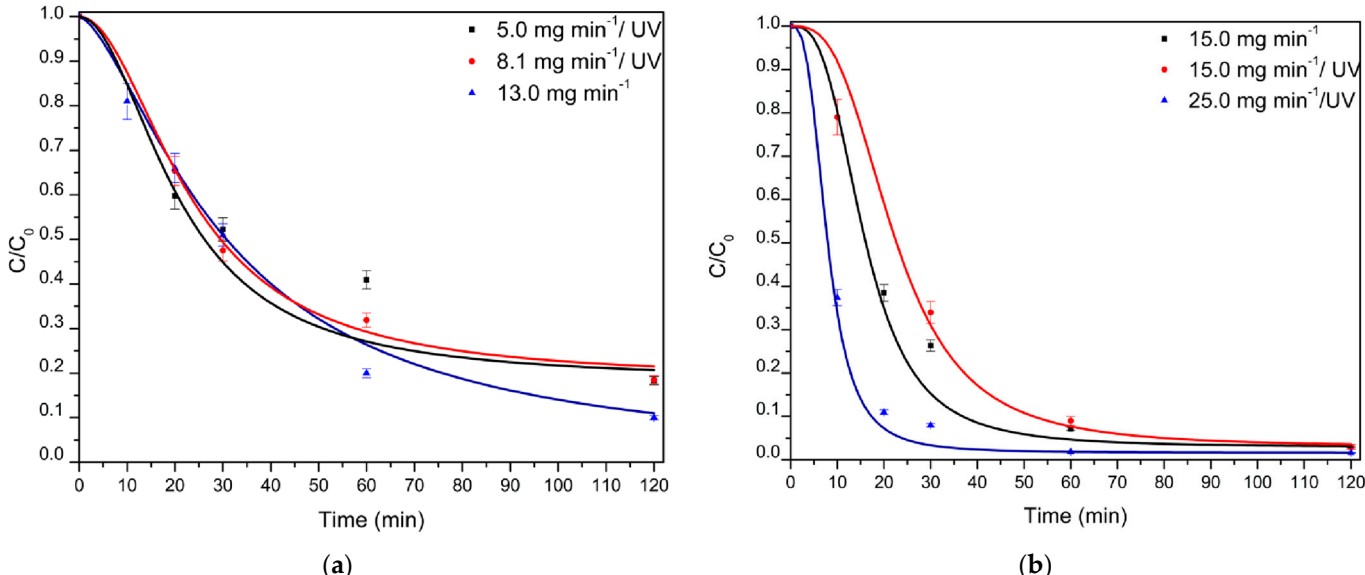

**Figure 8.** Normalized AMX concentration ($C/C_0$) abatement kinetics for photolytic (30-wa UV) and non-photolytic processes. pH 13.00 in all experiments. LP-Hg lamps (254 nm). (**a**) Ozone doses at 5.0, 8.13, and 13.0 mg min$^{-1}$. (**b**) 15.0 and 25.0 mg min$^{-1}$.

Table 1 presents a complete set of results of carbon and nitrogen analyses. Total carbon (TC), inorganic carbon (IC), total organic carbon (TOC, TOC = TC − IC), non-purgeable organic carbon (NPOC), and total nitrogen (TN) were analyzed.

The kinetic data extracted from the AOP experiments are presented in Table 2, with the respective correlation coefficient and calculated $E_{EO}$. The reaction rate constants are represented as pseudo-first-order kinetics, since the oxidation profiles fit reasonably well in experimental decays, as attested by the $R^2$ values.

**Table 1.** Analyses of carbon and nitrogen elements in the $O_3$-based oxidation of AMX. Total carbon (TC, mg L$^{-1}$), inorganic carbon (IC, mg L$^{-1}$), total organic carbon (TOC, mg L$^{-1}$), non-purgeable organic carbon (NPOC, mg L$^{-1}$), and total nitrogen (TN, mg L$^{-1}$).

| Process | Mass Flow (mg min$^{-1}$) | pH | Time (min) | TC (mg L$^{-1}$) | IC (mg L$^{-1}$) | TOC (mg L$^{-1}$) | NPOC (mg L$^{-1}$) | TN (mg L$^{-1}$) |
|---|---|---|---|---|---|---|---|---|
| $O_3$ | 8.1 | 9 | 0 | 183.3 | 0.95 | 182.3 | 190.00 | 9.23 |
| | | | 120 | 177.3 | 2.36 | 177.3 | 166.00 | 6.87 |
| | | 11 | 0 | 199.0 | 2.44 | 196.6 | 177.4 | 8.29 |
| | | | 120 | 204.3 | 8.88 | 195.8 | 183.50 | 9.27 |
| | 15.0 | 9 | 0 | 1094.0 | 6.90 | 1087.0 | 57.40 | 0.00 |
| | | | 120 | 719.2 | 14.58 | 704.6 | 3.50 | 41.00 |
| | | 11 | 0 | 169.2 | 1.34 | 167.8 | 175.90 | 8.57 |
| | | | 120 | 172.1 | 3.05 | 169.0 | 167.40 | 8.55 |
| | | 13 | 0 | 168.1 | 2.25 | 165.8 | 173.30 | 8.57 |
| | | | 120 | 165.2 | 4.48 | 160.7 | 144.90 | 8.52 |

**Table 1.** *Cont.*

| Process | Mass Flow (mg min$^{-1}$) | pH | Time (min) | TC (mg L$^{-1}$) | IC (mg L$^{-1}$) | TOC (mg L$^{-1}$) | NPOC (mg L$^{-1}$) | TN (mg L$^{-1}$) |
|---|---|---|---|---|---|---|---|---|
| O$_3$/UV | 15.0 | | | | | | | |
| | | 9 | 0 | 197.4 | 15.70 | 181.7 | 171.70 | 8.82 |
| | | | 120 | 189.1 | 20.33 | 169.0 | 170.50 | 8.39 |
| | | | 0 | 171.7 | 1.15 | 170.6 | 165.20 | 8.64 |
| | | | 120 | 112.6 | 2.90 | 109.7 | 111.30 | 6.10 |
| | | 11 | 0 | 172.6 | 0.97 | 171.6 | 179.60 | 9.19 |
| | | | 120 | 177.3 | 2.36 | 174.9 | 177.50 | 9.18 |
| | | | 0 | 212.7 | 19.49 | 193.2 | 177.40 | 8.88 |
| | | | 120 | 158.6 | 6.85 | 179.1 | 171.00 | 9.81 |
| | | 13 | 0 | 114.9 | 5.49 | 108.9 | 110.00 | 5.48 |
| | | | 120 | 162.3 | 6.65 | 155.6 | 146.60 | 7.77 |
| | | | 0 | 175.3 | 2.40 | 172.9 | 176.40 | 9.05 |
| | | | 120 | 157.5 | 1.12 | 156.4 | 152.20 | 9.46 |
| | | | 0 | 168.1 | 2.25 | 165.8 | 173.30 | 8.57 |
| | | | 120 | 165.2 | 4.48 | 160.7 | 144.90 | 8.52 |
| | | | 0 | 168.1 | 2.25 | 147.9 | 173.30 | 8.57 |
| | | | 120 | 165.2 | 4.48 | 133.6 | 144.90 | 8.52 |
| | | | 0 | 172.9 | 0.97 | 171.9 | 175.70 | 9.03 |
| | | | 120 | 166.4 | 1.16 | 165.2 | 169.70 | 8.25 |

**Table 2.** Apparent pseudo-first-order rate constants (min$^{-1}$) and electric energy per order ($E_{EO}$, kWh m$^{-3}$ order$^{-1}$) for the O$_3$-based AOPs.

| Process Condition | pH | k′ (min$^{-1}$) | $R^2$ | $E_{EO}$ (kWh m$^{-3}$ Order$^{-1}$) |
|---|---|---|---|---|
| O$_3$/UV 8.1 mg min$^{-1}$ | 9.0 | $(0.8 \pm 0.2) \times 10^{-2}$ | 0.925 | $176 \pm 40$ |
| | 11.0 | $(0.5 \pm 0.1) \times 10^{-2}$ | 0.886 | $282 \pm 51$ |
| | 13.0 | $(2.8 \pm 0.8) \times 10^{-2}$ | 0.976 | $50 \pm 13$ |
| O$_3$ 15.0 mg min$^{-1}$ | 9.0 | $(5.2 \pm 0.2) \times 10^{-2}$ | 0.999 | $20 \pm 1$ |
| | 11.0 | $(6.5 \pm 1.0) \times 10^{-2}$ | 0.986 | $16 \pm 2$ |
| | 13.0 | $(4.9 \pm 0.2) \times 10^{-2}$ | 1.000 | $21 \pm 1$ |
| O$_3$/UV 15.0 mg min$^{-1}$ | 9.0 | $(6.7 \pm 0.7) \times 10^{-2}$ | 0.992 | $21 \pm 2$ |
| | 11.0 | $(9.9 \pm 0.7) \times 10^{-2}$ | 0.997 | $14 \pm 1$ |
| | 13.0 | $(5.5 \pm 0.1) \times 10^{-2}$ | 1.000 | $26 \pm 1$ |

Lastly, Figure 9 shows the evolution of capital and operating costs (CAPEX and OPEX, respectively) for varying contributions of UV-irradiated volumes, f(UV). The simulations were carried out dividing the plant operating schedule into shifts (1, 2, or 3) and batches per shift (1 to 4), considering an average time of 40 min per batch and 8 h shifts. Table 3 summarizes the costs and equipment sizing required to operate a model WWTP based on O$_3$/UV at pH 13.0 in order to achieve an 80% conversion of AMX, assuming a daily throughput of 10 m$^3$ and [AMX]$_0$ of 10 mg L$^{-1}$.

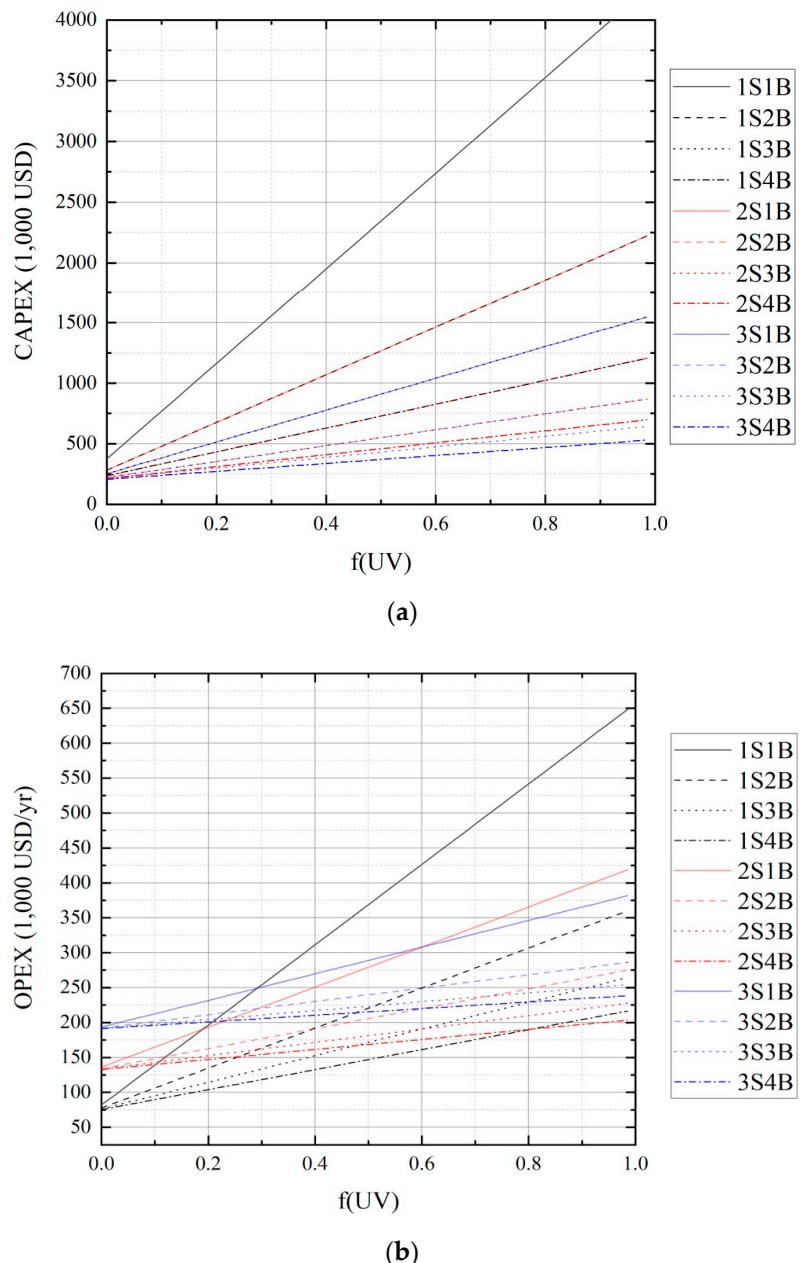

**Figure 9.** Impact of f(UV) on the (**a**) CAPEX and (**b**) OPEX for the three shift systems (1S, 2S, and 3S) with one to four operation batches (1B, 2B, 3B, and 4B) per shift.

**Table 3.** Detailed Cost Estimation and Operation Parameters.

| Equipment | Sizing | CAPEX (USD) | OPEX (10³ USD/yr) |
|---|---|---|---|
| Ozone generator | 800 g h$^{-1}$ | 15,720 | 934 |
| Ozonation tank | 2.5 m$^3$ | 17,300 | |
| H$_2$SO$_4$ tank | 2.0 m$^3$ | 7490 | 656 |
| NaOH tank | 4.5 m$^3$ | 11,460 | 4080 |
| ISBL factors | | | |

**Table 3.** *Cont.*

| Equipment | Sizing | CAPEX (USD) | OPEX ($10^3$ USD/yr) |
|---|---|---|---|
| Equipment erection | 0.3 | 15,850 | |
| Piping | 0.8 | 42,230 | |
| I&C | 0.3 | 15,850 | |
| Electrical | 0.2 | 10,560 | |
| Civil | 0.3 | 15,840 | |
| OSBL expenses | | 15,310 | |
| Engineering costs | | 50,510 | |
| Contingency reserve | | 16,840 | |
| Fixed costs | | | 69,570 |
| Variable costs | | | 57,880 |
| Expected batch time | | 44 min | |
| Cleaning/Idle time | | 3 h day$^{-1}$ | |

## 4. Discussion

Regarding the water matrix, most studies of degradation of pharmaceutical compounds by AOP deal with a standard analytical compound dissolved in ultrapure water, while real effluents or pharmaceutical formulation residues receive less attention. Our work differs substantially from this standard, since it is intended to degrade expired pharmaceutical formulations containing amoxicillin, hence dealing with real residues.

An important characteristic observed with samples of the commercial formulation is the large variability of initial concentration (Figure 4) for the same formulation amount. According to Figure 4 (right side), the distribution of $[AMX]_0$ is asymmetric with a median of ~98 mg L$^{-1}$. The asymmetry of the distribution of $[AMX]_0$ is positive with a small portion of the samples with concentrations above the median. The first quartile presents $[AMX]_0$ of ~85 mg L$^{-1}$, that is, at least 25% of the samples presented $[AMX]_0 = 85$ mg L$^{-1}$. The third quartile presents $[AMX]_0$ of ~160 mg L$^{-1}$, that is, 75% of the samples had concentrations up to $[AMX]_0 = 160$ mg L$^{-1}$. The minimum concentration of the studied samples was ~35 mg L$^{-1}$. The maximum concentration of the studied samples was ~260 mg L$^{-1}$. Despite having asymmetric dispersion, the number of outliers in the present study was insignificant, which is an indication of good chemical analytical and sample preparation qualities from our research group.

Figure 4 (left side) shows the $TOC_0$ distribution for the same group of samples analyzed in Figure 4 (right side). According to Figure 4, the distribution of $TOC_0$ is asymmetric with a median of ~172 mg L$^{-1}$. The asymmetry of the distribution of $TOC_0$ is negative. The first quartile presents $TOC_0$ of ~140 mg L$^{-1}$, that is, at least 25% of the samples presented $TOC_0 = 140$ mg L$^{-1}$. The third quartile presents $TOC_0$ of ~180 mg L$^{-1}$, that is, 75% of the samples had concentrations up to $TOC_0 = 180$ mg L$^{-1}$. The minimum $TOC_0$ concentration of the studied samples was ~125 mg L$^{-1}$. The maximum $TOC_0$ concentration of the studied samples was ~200 mg L$^{-1}$. Despite having asymmetric dispersion, the number of outliers in the present study was insignificant, which is an indication of good chemical analytical and sample preparation qualities from our research group.

There is a difference between the types of asymmetries between the initial AMX, $[AMX]_0$ (Figure 4, right) and total organic carbon, $TOC_0$ (Figure 4, left) concentration results. It is necessary to consider, in addition to the asymmetric variability of the antibiotic, the variability of each component that makes up the commercial formulation of the drug used in the current study. If the $TOC_0$ variability depended solely on the $[AMX]_0$ present in the formulation, the $TOC_0$ asymmetry would follow the $[AMX]_0$ asymmetry.

Preliminary studies were also carried out in acid media (pH 3 and 5). However, in a basic medium, amoxicillin is more soluble, and no precipitate was observed. All tests of ozone-based advanced oxidation processes in acidic conditions resulted in formation of precipitates. All considerations in the present study are based on experiments at basic-pH values (Figures 5–8).

The results shown in Figure 5 correspond to the AMX degradation by hydrolysis in basic aqueous media, an irreversible chemical reaction. In high pH, the unstable β-lactam ring suffers hydrolysis [33]. The β-lactam ring is highly tensioned and thus easy to be broken. In other words, there was the opening of the β-lactam ring with the AMX hydrolysis [34] in basic media. According to Aryee et al. [33], the opening of the β-lactam ring is the main reason for the low detection of AMX in surface waters compared with other antibiotics. Moreover, higher concentrations of the hydroxide anion ($HO^-$) promote higher degradation of AMX. At pH 9, a degradation of ~10% was observed after 30 min; at pH 11, ~20% of AMX degradation was achieved at the same time, whereas at pH 13, a degradation of ~40% was observed after 20 min.

AMX degradation studies by advanced oxidation processes have pointed to hydrolysis mechanisms. Other authors disregard in their analyses the hydrolysis mechanisms [27,35–38]. However, many authors are limited to the qualitative evaluation of such an important abiotic AMX degradation mechanism [39,40]. Andreozzi et al. [41] studied the removal of antibiotics in water by ozonation. The authors investigated the degradation in a gas–liquid semi-continuous reactor at 25 °C, feeding it at a volumetric flow rate of 36 L $h^{-1}$. At pH = 5.5, Andreozzi et al. [41] achieved AMX degradation >90% in 4 min of reaction and mineralization of only ~18% in 20 min of reaction. The hydrolysis evaluation was only qualitative, indicating the formation of 2-amino-2-(p-hydroxyphenyl)acetic acid in aqueous medium. Cao et al. [35] studied the degradation of several antibiotics, including AMX, in aqueous media with catalysts and ozone. Cao et al. [35] were able to remove 100% of the antibiotics with a residence time of 71 s. However, they neglected the contribution and possible hydrolysis involved in the chemical processes.

The region of pH (Figures 5–8) corresponds to the negatively charged amoxicillin molecules ($AMX^{2-}$). The deprotonated form of amoxicillin ($AMX^{2-}$) was more susceptible to ozone attack. Moreover, the decomposition rate of ozone depended on the pH. The direct attack on organic compounds by ozone occurred under basic conditions. At a high pH value [42], ozone was partially decomposed to non-selective $^\bullet OH$ [Equation (7)], which could easily attack the molecule of AMX.

$$2O_3 + OH^- \rightarrow \ OH + O_2^- + 2O_2 \tag{7}$$

Ozone has a high oxidizing ability (2.07 V) and can oxidize several recalcitrant organic pollutants. During the production of ozone, hydroxyl radicals ($^\bullet OH$), superoxide radicals ($O_2^{\bullet -}$), and hydroperoxide radicals ($HO_2^\bullet$) can be formed by chain reactions [43]. Among radicals, hydroxyl radicals are the most effective for the degradation of organic pollutants and are more beneficial in the degradation of AMX than direct oxidation of $O_3$. In alkaline conditions (Figures 6–8), more $O_3$ may be transformed into $^\bullet OH$, and indirect oxidation is the main path. The ozone concentration has an important influence on the degradation of AMX. In fact, according to the data shown in Figures 6–8, ozone concentration is the main parameter of AMX degradation. The mass transfer rate and the volumetric mass transfer coefficient of ozone increase with ozone concentration. More ozone can be absorbed and reacts with AMX molecules, finally improving the degradation of amoxicillin.

In ozonation, only those functional groups especially reactive to an electrophilic reagent can be easily oxidized by $O_3$. In the case of amoxicillin, the reaction with $O_3$ occurs in the phenolic group and in the non-protonated amino group [44]. As the experiments were carried out at pH greater than 7.49 ($pKa_2$), the non-protonated amino group prevailed over its protonated form. However, only the $^\bullet OH$ radicals produced by the partial decomposition of dissolved $O_3$ may attack those less reactive functional groups, such as aliphatic hydrocarbon and carboxylic groups [44] present in the amoxicillin structure.

Flores-Payán et al. [45] studied a computational model for a reactor used in advanced water treatment to represent, simulate, and predict the mass transfer process of ozone in water (gas–liquid). For the overall mass transfer coefficient ($k_{La}$), it was reduced from $5.30 \times 10^{-3}$ $s^{-1}$ at pH = 7 to $3.17 \times 10^{-3}$ $s^{-1}$ at pH = 11. On the other hand, the self-decomposition ($K_c$) of $O_3$ to $^\bullet OH$ increased from $3.4 \times 10^{-3}$ $s^{-1}$ at pH = 7 to $5.59 \times 10^{-3}$ $s^{-1}$

at pH = 11. At high pH, as used in the present work, the $O_3$ mass transfer from the gas phase to the liquid phase tends to decrease, while the production of $^\bullet$OH by the self-decomposition mechanism tends to increase. At the same time occurred a direct $O_3$ attack in electrophilic groups (phenolic and non-protonated amino groups) and a non-specific attack provided by $^\bullet$OH.

In terms of chemical mechanisms of the AMX degradation, the UV process (Figures 6–8) involved in the direct photolysis of AMX molecules and generation of additional $^\bullet$OH by $O_3$ is less representative. The pharmaceutical formulation contains various organic and inorganic substances that absorb most of the UV emitted by the low-pressure mercury lamps ($\lambda$ = 254 nm). According to the pharmaceutical manufacturer, the amoxicillin formulation contains sodium benzoate, sodium citrate dihydrate, a cherry flavor substance, a strawberry flavor substance, silicon dioxide, xanthan gum, disodium erythrosine red, and sucrose. Such substances described by the manufacturer can compete with the AMX photolytic degradation pathway and with the indirect ozonation mechanism (Equations (8) and (9)).

$$O_3 + UV \rightarrow O_2 + O\left(^1D\right) H_2O \rightarrow H_2O_2 \tag{8}$$

$$H_2O_2 \rightarrow 2\bullet OH \tag{9}$$

When exposed to UV light (<300 nm), ozone is photolyzed into oxygen and an excited oxygen atom [$O(^1D)$]. In a further step, in water, $O(^1D)$ adds to $H_2O$ to produce hydrogen peroxide. There is not a direct production of $^\bullet$OH as occurs in the gaseous phase in the presence of water [46].

The ozonation combined with UV radiation ($O_3$/UV) is initiated by the photolysis of ozone and followed by the production of $^\bullet$OH radicals by the reaction of $O^\bullet$ with water [47], as described by Equations (10)–(12).

$$O_3 + UV \rightarrow O_2 + O^\bullet \tag{10}$$

$$O^\bullet + H_2O \rightarrow 2OH^\bullet \tag{11}$$

$$2O^\bullet + H_2 \rightarrow HO^\bullet + HO^\bullet \rightarrow H_2O_2 \tag{12}$$

Mechanisms of hydrolysis, photolysis, and direct/indirect ozonation in basic media were able to achieve a high degree of AMX degradation. However, most experiments were unable to achieve mineralization of AMX, and the pharmaceutical formulation content dissolved into the water. Non-purgeable organic carbon (NPOC, Table 1) represents all the organic carbon after the sample has been acidified (HCl) and purged with high purified air to remove volatiles. In such sample preparation, volatile organic content, light hydrocarbons, and solvents are lost during the purging process. It can be noticed that there is no significance between TOC and NPOC data (Table 1). In such a situation, it can be concluded that the pharmaceutical residues dissolved in water presented no volatile organic substances. Volatile substances that could be purged by the inlet gaseous stream applied into each ozonation-based process. Air stripping is not a viable mechanism in the present work, and the AMX degradation was provided by mechanisms of hydrolysis, photolysis, and direct/indirect ozonation in basic media.

Inorganic carbon (IC, Table 1) represents bicarbonate and carbonate ions in basic aqueous solutions. At pH > 11 it is expected that only carbonates exist as inorganic carbon. In many ozonation-based experiments, it was found increasing in the IC after 120 min. For instance, at pH 9.00 and 15.00 mg $O_3$ min$^{-1}$, IC was doubled from 6.90 mg L$^{-1}$ to 14.58 mg L$^{-1}$. Increase in IC values is an indication of mineralization of the organic carbon. The majority of data show IC in the reaction AMX solutions. Moreover, IC anions do not react directly with ozone molecules, but bicarbonate (Equation (13)) and carbonate (Equation (14)) may act as hydroxyl radical scavengers, increasing complexity in the reaction media. On the other hand, the resulting anion-radical $CO_3^{\bullet-}$, with potential 1.78 V (pH 7), is selective and able to react with electron-rich compounds [48]. AMX is a molecule with

electron-rich organic groups such as phenols, nitrogen, and sulfur [48]. $CO_3^{\bullet-}$ may react with high second-order rate constant values ($10^5$ to $10^9$ $M^{-1}$ $s^{-1}$) [48].

$$HCO_3^- + HO^\bullet \rightarrow CO_3^{\bullet-} + H_2O \tag{13}$$

$$CO_3^{2-} + HO^\bullet \rightarrow CO_3^{\bullet-} + OH^- \tag{14}$$

Total nitrogen (TN, Table 1) represents organic and inorganic nitrogen-based compounds, and the applied methods do not differentiate, for example, nitrite/nitrate from organic nitrogen of AMX molecules. Nitrites and nitrates are more likely to be the final transformation products, as there is no pattern of decreasing TN values (Table 1).

Total organic carbon (TOC, Table 1) represents the sum of organic carbon in the solutions. It also represents organic carbon from the transformation products. Decreasing of TOC values represents the mineralization of the organic carbon into bicarbonate and carbonate ions at basic media. It was found to decrease the AMX concentration by spectrophotometric measures (Figures 5–8), and it was found to increasing the IC (Table 1). Those facts indicate degradation of the organic compounds, but without enough energy to achieve a high level of mineralization.

*4.1. Kinetic Modeling and the Electric Energy per Order ($E_{EO}$)*

As can be seen from Table 2, the combination of ozone and UV irradiation promoted a synergistic effect, increasing by ca. 67% the apparent reaction rate constant at the photochemical reactor; and the process is very sensitive to the initial pH.

Li et al. [5] studied the degradation of methyl orange (MO), amoxicillin (AMX), and 3-chlorophenol (3-CP) in water by photolytic processes. Li et al. [5] used a 10 W UV black light lamp ($\lambda_{max}$: 365 nm) as the irradiation source and anatase $TiO_2$ as a photocatalyst. The photocatalytic degradation of MO, AMX, and 3-CP was performed by circulating each solution (25 mg $L^{-1}$) through CMCPR under UV irradiation (7.5 W $m^{-2}$) for 300 min. The percent reduction of 3-CP reached 100% in 180 min, while that of MO and AMX was only 80.2% and 74.6%, respectively, at the end of the 300-min UV irradiation. For AMX, the apparent rate constant was 0.0042 $min^{-1}$, and the $E_{EO}$ was $7.31 \times 10^4$ kW h $m^{-3}$ $order^{-1}$. In relation to our work, Li et al. [5] used pure antibiotics dissolved in water, but Table 2 shows better results. In terms of chemical kinetics, Table 2 shows kinetic results up to 23 times higher than those presented by Li et al. [5]. In terms of $E_{EO}$, Li et al. [5] reported energy consumption results up to 5200 times higher than ours (Table 2).

Alaton et al. [37] combined chemical and biological oxidation to degrade 1.0 L penicillin formulation effluent. The ingredients of the penicillin formulation effluent were amoxicillin trihydrate (penicillin active substance), potassium clavulanate (assistant active substance), croscarmellose sodium (flavor), HDK-N20 (binding agent), sodium stearyl fumarate (additive), and Avicel pH 11.2 (buffering agent). The average Dissolved Organic Carbon (DOC) of the penicillin formulation effluent was 450 mg $L^{-1}$. The authors did not measure the AMX concentration. For the 1 h ozonation process, ozone dose of 2500 mg $L^{-1}$ $h^{-1}$, the best COD removal was 56% at pH 12.0 with first-order COD removal rate constant of 0.0134 $min^{-1}$. Alaton et al. [37] did not present the electric energy order, but considering ozone generator rate power of 80 W, and effluent volume of 1 L, the ozonation energy consumption was $E_{EO}$ ~ 229 kWh $m^{-3}$ $order^{-1}$. In relation to our work, Alaton et al. [37] used different pharmaceutical formulations, but a quite similar effluent treatment process when compared with ours. Table 2 shows kinetic results up to seven times higher than those presented by Alaton et al. [37].

Elmolla and Chaudhuri [49] studied photocatalytic degradation of three antibiotics in water: amoxicillin (AMX), ampicillin (AMP), and cloxacillin (CLX). Dissolution of pure antibiotics in distilled water was the basis for the $UV/TiO_2$ and $UV/H_2O_2/TiO_2$ photocatalysis processes. The aqueous solution characteristics were AMX, AMP, and CLX concentrations of 104, 105, and 103 mg $L^{-1}$, respectively, pH ~5, chemical oxygen demand (COD) of 520 mg $L^{-1}$, and dissolved organic carbon (DOC) of 145 mg $L^{-1}$. Additionally, the UV system was based

on a UVA source lamp with nominal power of 6 W. Elmolla and Chaudhuri [49] reported a pseudo-first order kinetics and the rate constants ($k$) equal to 0.007, 0.003, and 0.029 min$^{-1}$ for amoxicillin, ampicillin, and cloxacillin, respectively. Addition of $H_2O_2$ at natural pH ~5 and $TiO_2$ 1.0 g L$^{-1}$ resulted in complete degradation of amoxicillin, ampicillin, and cloxacillin in 30 min. Based on the data provided by Elmolla and Chaudhuri [49], the electric energy $E_{EO}$ were ~11, ~26, and ~3 kWh m$^{-3}$ order$^{-1}$ for AMX, AMP, and CLX, respectively. Both energy consumption and degradation kinetics from Elmolla and Chaudhuri [49] present results in the same order of magnitude of the present work (Table 2). In relation to our work, Elmolla and Chaudhuri [49] used pure antibiotics dissolved in water, but the results are similar. It is necessary to consider that the present work deals with a much more complex water matrix, making it unnecessary to separate a catalyst after the treatment process or even apply UV radiation, as can be seen in Table 2.

Mohammadi et al. [50] studied degradation of amoxicillin trihydrate by a $Sn/Zn/TiO_2$ photocatalyst. It was prepared with a 20 mg L$^{-1}$ antibiotic with pure amoxicillin trihydrate in purified water. The batch photoreactor was equipped with 36 W black light lamp ($\lambda_{max}$ = 365 nm). In each run, 400 mg L$^{-1}$ of prepared photocatalyst was dispersed in 100 mL water. Mohammadi et al. [50] reported electric energy $E_{EO}$ for four catalysts in the batch photoreactor: (i) $TiO_2$, 40.66 kWh m$^{-3}$ order$^{-1}$; (ii) $Zn/TiO_2$, 28.36 kWh m$^{-3}$ order$^{-1}$; (iii) $Sn/TiO_2$, 21.71 kWh m$^{-3}$ order$^{-1}$; and (iv) $Sn/Zn/TiO_2$, 10.15 kWh m$^{-3}$ order$^{-1}$. The energy consumption from Mohammadi et al. [50] is on the same order of magnitude as the present work (Table 2). However, the $E_{EO}$ results (Table 2) are based on the degradation of AMX in higher concentrations and in the presence of several additives (sodium benzoate, sodium citrate dihydrate, a cherry flavor substance, a strawberry flavor substance, silicon dioxide, xanthan gum, disodium erythrosine red, and sucrose).

### 4.2. Scaling up and Implementation Costs

The practical implementation of $O_3$-based treatment technologies involves the acquisition of specialized equipment (ozone generator and UV systems, for $O_3$/UV processes), in addition to process consumables, including acids and bases for upstream and downstream pH adjustment. Considering the results observed in previous sections, it seems that the most efficient operating conditions involve the use of $O_3$ with or without irradiation at pH 13, controlled by the addition of NaOH. Therefore, our scaling-up calculations for batch processing were carried out considering the kinetics obtained in this scenario. Figure 9 shows that CAPEX costs can be reduced by 10-fold just by changing the operating system: this results from the possibility of using smaller (and cheaper) equipment when the daily throughput can be processed in more batches. Conversely, OPEX costs increase with increasing the number of shifts, a consequence of the necessity of more labor hours and utilities consumption, and all costs involved. An analysis of CAPEX and OPEX curves (Figure 9) indicates that operation in four batches is the most cost-effective in a single shift. Moreover, the curves show that increasing the fraction of UV-irradiated volume, f(UV) = $V_{UV}/V_{system}$, increases both capital and operation costs substantially. Ozonation alone, in basic pH conditions, is the most cost-effective course of action for scaling up the AMX degradation process when adopting a four-batch single-shift operation regime without any UV reactor (f(UV) = 0). This is based on the cost analyses (CAPEX, OPEX), the energy consumption $E_{EO}$ results, and supported by our experimental evidence that has shown that ozonation alone (Figures 6–8) is sufficient to achieve a desirable AMX degradation. Table 3 summarizes the costs and equipment sizing required in this condition. The amount of NaOH and $H_2SO_4$ used in the design of the tanks and calculation of OPEX were proportional to the amount of solution used in the semi-batch experiments to promote pH adjustment.

## 5. Conclusions

From the chemistry point of view, effective degradation of amoxicillin was achieved using UV–$O_3$-based AOPs. Hydrolysis, direct ozonation, and radical reactions are the main mechanisms for the AMX degradation in the studied effluent at high pH. In a complex

effluent, with a high concentration of AMX and several non-volatile organic and inorganic excipients, it was possible to obtain a high percentage of AMX degradation. However, the transformation products proved to be recalcitrant to the AOPs studied. In terms of kinetics and energy consumption, the proposed process was even more efficient than previous studies in the literature involving simple effluent with the dissolution of pure AMX in water. Despite the $O_3/UV$ process being more efficient than the $O_3$ process in basic media, the CAPEX and OPEX analysis showed that the application of UV lamps is economically unfeasible. Considering the energy consumption and lamp replacement cost, the best option for scaling up is to avoid a system with a UV reactor, relying on an ozone generator. The electric energy per order ($E_{EO}$) results also indicate, as the best option, a system with no UV reactor. $E_{EO}$ of the present work showed a lower energy consumption system than previous papers of degradation of pure amoxicillin dissolved in water.

**Supplementary Materials:** The following supporting information can be downloaded at: https://www.mdpi.com/article/10.3390/w14203198/s1, Figure S1: Microspecies #1, Figure S2: Microspecies #2, Figure S3: Microspecies #3, Figure S4: Microspecies #4, Figure S5: Microspecies #5, Figure S6: Microspecies #6, Figure S7: Microspecies #7, Figure S8: Microspecies #8.

**Author Contributions:** Conceptualization, A.L.d.C.P.; methodology, B.S.S., M.C.B.R., B.R., and A.L.d.C.P.; validation, B.S.S., M.C.B.R., B.R. and A.L.d.C.P.; formal analysis, B.S.S., M.C.B.R., B.R., and A.L.d.C.P.; investigation, B.S.S. and M.C.B.R.; resources, A.L.d.C.P.; writing—original draft preparation, B.S.S., M.C.B.R., B.R. and A.L.d.C.P.; writing—review and editing, B.S.S. and A.L.d.C.P.; visualization, B.S.S. and M.C.B.R.; supervision, A.L.d.C.P.; project administration, A.L.d.C.P.; funding acquisition, A.L.d.C.P. All authors have read and agreed to the published version of the manuscript.

**Funding:** The research was funded by the São Paulo Research Foundation (FAPESP), Grants 2018/05698-0 (Regular Research Grant), 2018/17913-2 (Multi-user Equipments), 2019/09991-6 (Scientific Initiation), and 2021/03446-6 (Scientific Initiation).

**Institutional Review Board Statement:** Not applicable.

**Informed Consent Statement:** Not applicable.

**Data Availability Statement:** The authors confirm that the data supporting the findings of this study are available within the article.

**Conflicts of Interest:** The authors declare no conflict of interest.

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
