# Peer review of "Removal of Amoxicillin from Processing Wastewater by Ozonation and UV-Aided Ozonation: Kinetic and Economic Comparative Study"

_water, doi:10.3390/w14203198_

Round 1

Reviewer 1 Report

Prevention of pollution of antibiotics like amoxicillin (AMX) is important. In this research, AMX treatment by using ozonation and UV-aided ozonation was investigated, and the process was modeled, the cost was assessed. It can be accepted after revision. The specific comments are as follows.

(1) AMX degradation is efficient at a higher pH 13. The waste water needs to be neutralized and the resulted salt might need to be removed before discharging if the salt content exceeds the limit of the government regulation. The cost of above operations might be much higher than that of electricity used by UV.

(2) A relative simple model (pseudo first order model) was used, and some important variables like pH and O3/UV were not included in the model as the independent variables, which were useful for obtaining the optimal operation conditions for cost calculation.

(3) Delete lines 31-39 of page 1 in the Introduction section.

(4) Line 75, page 3, “2” in “(H2O2) ” should be subscript.

(5) Delete “in” in line 109, page 3.

Reviewer 2 Report

This study investigated the degradation of amoxicillin using UV-O3-based advanced oxidation technology for wastewater treatment under the high pH. They investigated the effects of pH, dosage on the TOC,TN removal and amoxicillin removal, yielding some possible application for wastewater treatment. Overall, the work is interesting. However, there still some major comments that need to be addressed before consideration for publication.

1.     The figures are poorly prepared, kindly redrawn the figures following the standards format.

2.     Introduction, the biological treatment methods need to be updated by citing Science of The Total Environment,2022,853, 158424.

3.     What is the kind of wastewater? AMX concentration of 500 ppm is too high. In addition, in the stimulated wastewater, any kinds of other organic matter?

4.     TOC and TN methods can be briefly by citing references.

5.     In the tank in figure 3, how to ensure the similar performance of O3 oxidation in the field scale application and the related cost analysis?

6.     Fig.4, TOC and AMX concentration was similar without other organic matter?

7.     Fig. 9, can the lab-scale experiment directly used for capital cost and operating cost evaluation?

Reviewer 3 Report

The manuscript related to the removal of Amoxicillin from Processing Wastewater by Ozonation and UV-aided Ozonation: Kinetic and Economic Comparative Study is interesting and within the scope of Water journal.

But after careful evaluation, I can find manuscript needs to be resubmitted

The author has mentioned the following sentences at the starting of the introduction section. “to improve the script. The introduction should briefly place the study in a broad context and highlight why it is important. It should define the purpose of the work and its significance. The current state of the research field should be carefully reviewed and key publications cited. Please highlight controversial and diverging hypotheses when necessary. Finally, briefly mention the main aim of the work and highlight the principal conclusions. As far as possible, please keep the introduction comprehensible to scientists outside your particular field of research. References should be numbered in order of appearance and indicated by a numeral or numerals in square brackets—e.g., [1] or [2,3], or [4–6]. See the end of the document for further details on references”. Is this the introduction or the comments of reviewer. This is totally unaccepted. This confirm author have not seriously prepared the manuscript. Under introduction section author have mention reviewer comments.

Page 2 author have mention “Antibiotic consumption in Brazil is approximately 22.75 DDD, 52 higher than other countries in the Americas such as Bolivia (19.57), Paraguay (19.38), 53 Canada (17.05), Costa Rica (14.18) and Peru (10.26) (Li et al., 2021).” Why the author has not followed the reference style? Some place numerical and someplace with authors and year. Not acceptable.

Kindly go through the comments and resubmit the manuscript for evaluation.

Round 2

Reviewer 2 Report

The comments have been addressed, thus can be accepted for possible publication.

Reviewer 3 Report

The authors have satisfactorily revised the manuscript.